# Quantitative Chemical Exchange Saturation Transfer Imaging of Amide Proton Transfer Differentiates between Cerebellopontine Angle Schwannoma and Meningioma: Preliminary Results

**DOI:** 10.3390/ijms231710187

**Published:** 2022-09-05

**Authors:** Hirofumi Koike, Minoru Morikawa, Hideki Ishimaru, Reiko Ideguchi, Masataka Uetani, Takeshi Hiu, Takayuki Matsuo, Mitsuharu Miyoshi

**Affiliations:** 1Department of Radiology, Nagasaki University Graduate School of Biomedical Sciences, 1-7-1 Sakamoto, Nagasaki 852-8501, Japan; 2Department of Radiology, Nagasaki University Hospital, 1-7-1 Sakamoto, Nagasaki 852-8501, Japan; 3Department of Radioisotope Medicine, Nagasaki University Graduate School of Biomedical Sciences, 1-7-1 Sakamoto, Nagasaki 852-8588, Japan; 4Department of Neurosurgery, Nagasaki University Graduate School of Biomedical Sciences, 1-7-1 Sakamoto, Nagasaki 852-8501, Japan; 5MR Application and Workflow, GE Healthcare Japan, Hino, Tokyo 191-8503, Japan

**Keywords:** magnetic resonance imaging, amide proton transfer imaging, schwannomas, meningiomas

## Abstract

Vestibular schwannomas are the most common tumor at the common cerebellopontine angle, followed by meningiomas. Differentiation of these tumors is critical because of the different surgical approaches required for treatment. Recent studies have demonstrated the utility of amide proton transfer (APT)-chemical exchange saturation transfer (CEST) imaging in evaluating malignant brain tumors. However, APT imaging has not been applied in benign tumors. Here, we explored the potential of APT in differentiating between schwannomas and meningiomas at the cerebellopontine angle. We retrospectively evaluated nine patients with schwannoma and nine patients with meningioma who underwent APT-CEST MRI from November 2020 to April 2022 pre-operation. All 18 tumors were histologically diagnosed. There was a significant difference in magnetization transfer ratio asymmetry (MTR_asym_) values (0.033 ± 0.012 vs. 0.021 ± 0.004; *p* = 0.007) between the schwannoma and meningioma groups. Receiver operative curve analysis showed that MTR_asym_ values clearly differentiated between the schwannoma and meningioma groups. At an MTR_asym_ value threshold of 0.024, the diagnostic sensitivity, specificity, positive predictive value, and negative predictive values for MTR_asym_ were 88.9%, 77.8%, 80.0%, and 87.5%, respectively. Our results demonstrated the ability of MTR_asym_ values on APT-CEST imaging to discriminate patients with schwannomas from patients with meningiomas.

## 1. Introduction

Chemical exchange saturation transfer (CEST) is an important magnetic resonance imaging (MRI) method that draws contrast from proton exchange between labeled protons in solutes and free bulk water protons [1,2]. Amide proton transfer (APT) imaging is the most often reported CEST technique and is based on amide protons associated with mobile proteins and peptides resonating at 3.5 ppm downfield from water [3,4]. This MRI technique has shown potential clinical utility for differentiating radiation necrosis from tumor recurrence or progression and differentiating high-grade from low-grade gliomas [5,6,7]. Although the source of APT signal intensity (SI) in brain tumors is unknown, previous studies have reported that the high cellular content of proteins and peptides in malignant tumors with increased cellularity can cause increased SI [6,7,8,9,10,11]. However, there are few reports on APT SI in benign brain tumors.

Approximately 10% of all intracranial tumors arise at the cerebellopontine angle (CPA). Most CPA tumors (80%) are vestibular schwannoma (VS) [12,13], followed by meningioma, which constitute 10–15% of tumors at the CPA [14]. VSs and meningiomas at the CPA are generally benign [15,16]. Preoperative differentiation between these two tumors is important because patients with meningioma have a better surgical prognosis and less chance of facial or vestibulocochlear nerve damage than patients with schwannomas [17,18].

Although conventional MRI techniques facilitate the preoperative diagnosis between schwannomas and meningiomas, some cases remain difficult to distinguish.

One study reported that APT-CEST imaging can differentiate between benign and atypical meningioma [19], while a second reported that APT-CEST imaging can differentiate between growing meningiomas and non-growing meningiomas [20]. However, to the best of our knowledge, no studies have examined the use of APT-CEST imaging for schwannomas.

The objective of the present study was to investigate the differences in APT signals between schwannomas and meningiomas at the CPA, comparing APT findings with preoperative MRI findings. We also aimed to determine the value of APT imaging for differentiating between these tumors.

## 2. Results

### 2.1. Clinical Characteristics and Imaging Findings of Schwannomas and Meningiomas

Among the 18 patients included in this study, nine patients (50%) had schwannomas, and nine patients (50%) had meningiomas (Table 1).

There was no significant difference between the two groups in age (*p* = 0.506), sex (*p* = 1.000), mean maximum tumor diameter (MTD) value (*p* = 0.173), T2 hyperintensity (*p* = 0.058), peritumoral brain edema (*p* = 0.599), irregular tumor margin (*p* = 0.137), heterogeneous enhancement (*p* = 0.058), capsular enhancement (*p* = 0.599), progress to the internal auditory canal (*p* = 0.629), and eccentricity of tumor compared to nerve (*p* = 0.058). Patients with tumors showing T2 hyperintensity and those with tumors showing heterogeneous enhancement were the same. Patients with tumors showing peritumoral brain edema and those with tumors showing capsular enhancement were also the same.

However, the T2 hyperintensity, heterogeneous enhancement, and eccentricity of tumor compared to nerve showed near significant differences between the two groups (*p* = 0.058). We speculate that the difference may be significant with more patients. A significant difference in mean magnetization transfer ratio asymmetry (MTR_asym_) (*p* = 0.007) and dural tail sign (*p* = 0.003) was found between the schwannoma group and the meningioma group.

Representative images are shown in Figure 1, Figure 2 and Figure 3.

### 2.2. Schwannomas with T2 Hyperintensity and T2 Low Intensity

Among the nine patients with schwannomas, six patients (66.7%) had schwannomas with T2 hyperintensity, and three patients (33.3%) had schwannomas with T2 low intensity (Table 2).

There was no significant difference between the two groups in age (*p* = 0.267), gender (*p* = 0.571), mean MTD value (*p* = 0.121), peritumoral brain edema (*p* = 0.571), irregular tumor margin (*p* = 0.571), capsular enhancement (*p* = 0.343), dural tail sign (*p* = N.S.), and progress to the internal auditory canal (*p* = 1.000). There was a significant difference in mean MTR_asym_ (*p* = 0.014) and heterogeneous enhancement (*p* = 0.003) between the schwannoma with T2 hyperintensity group and the schwannoma with T2 low-intensity group.

### 2.3. Pathological Findings

The nine resected schwannomas were diagnosed as benign and had no malignant findings. Among the nine meningiomas, six meningiomas were diagnosed as transitional meningiomas (World Health Organization (WHO) grade I), two meningiomas were diagnosed as meningothelial meningiomas (WHO grade I), and one was diagnosed as fibrous meningioma (WHO grade I).

### 2.4. ROC Analysis in Patients in the Schwannoma and Meningioma Groups

Receiver operating characteristic (ROC) analyses demonstrated moderate discriminatory power (AUC = 0.73) for mean MTR_asym_ to differentiate between patients in the schwannoma and meningioma groups. When a cut-off value <0.024 was used as the threshold for diagnosis, the sensitivity, specificity, positive predictive value, and negative predictive value were 88.9%, 77.8%, 80.0%, and 87.5%, respectively (Figure 4).

### 2.5. Interobserver Agreement

Interobserver agreement regarding MTR_asym_ and MTD was excellent with intraclass coefficient (ICC) values of 0.941 [95% confidence interval (CI), 0.850–0.978] and 0.969 [95% CI, 0.920–0.988]). Interobserver agreement on specific conventional imaging features was also good for all parameters (k values: 0.655–1.000).

## 3. Discussion

MRI has been applied for the diagnosis of CPA tumors. VS is the most common tumor in the CPA, accounting for 70–80% of all CPA masses [21]. Three different MRI appearances of the tumoral tissue are reported from T2-weighted images and postgadolinium T1-weighted images: homogeneous (50–60%), heterogeneous (30–40%), and cystic (5–15%) [22,23].

Histologically, VS is frequently a biphasic lesion with alternating areas of two types of patterns: the Antoni type A pattern, a dense tissue made of elongated bipolar cells with club-shaped nuclei disposed of in fascicles, and the Antoni type B pattern, relatively less cellular and loose textured tissue composed of uniform, small, somewhat stellate cells, with round condensed nuclei and indistinct cytoplasm [24].

The size of the VS is correlated to the appearance of the signal and the gadolinium uptake and the histological Antoni subtype: small VSs are usually homogeneous and histologically composed of Antoni type A pattern, while heterogeneous and cystic VSs are larger and include Antoni B pattern or a mix of type A and B [23]. VSs that are larger than 25 mm in diameter are heterogeneous because of the occurrence of additional cystic or necrotic components [25].

Meningioma is the second most frequent lesion in the CPA after VS, representing 10–15% of all tumors in this location [26]. MRI of meningioma clearly depicts a broad-based dural hemispheric or oval lesion attached to the petrous dura matter or the inferior aspect of the tentorium. Meningiomas are usually isointense with cortex on all sequences and strongly enhance after contrast injection, often homogeneously. Though not specific to meningiomas [27], the enhancement of the thickened peritumoral dura, the so-called dural tail sign, is particularly frequent with meningiomas and should suggest meningioma diagnosis when observed. In this study, the dural tail sign was found only in the meningioma group.

One report showed that proton MR spectroscopy may help in distinguishing schwannomas from meningiomas. It depicts a prominent myo-inositol peak in schwannomas, whereas the alanine found in meningiomas is absent in schwannomas [28].

Several studies have evaluated APT imaging for evaluating brain tumors [9,29,30]. One report showed the feasibility of APT imaging for grading meningiomas [19], and another study also investigated the feasibility of APT imaging to differentiate between growing meningiomas and non-growing meningiomas [20]. However, to the best of our knowledge, our results are the first to describe the ability of APT imaging to differentiate between schwannomas and meningiomas at the CPA.

Recent studies on gliomas and meningiomas have demonstrated the positive correlation of APT SI with cell proliferation index [6,10,19]. These results suggested that high-grade tumors, which show higher proliferation, have higher densities and numbers of cells (with higher concentrations of intracellular proteins and peptides) than low-grade tumors. In contrast, low-grade tumors tend to show low APT SI.

In this study, we found a significant difference in the mean MTR_asym_ between the patients in the schwannoma group and those in the meningioma group. The mean MTR_asym_ in the schwannoma group was higher than that of the meningioma group, and ROC analyses demonstrated moderate discriminatory power for mean MTR_asym_ to differentiate between patients in the schwannoma and meningioma groups. We also found a significant difference in mean MTR_asym_ between schwannomas with T2 hyperintensity and schwannomas with low intensity; the mean MTR_asym_ in schwannomas with T2 hyperintensity was higher than that of schwannomas with low intensity. There are no previous reports of APT imaging for schwannomas, and the association between Antoni type B and APT SI has been unknown. A previous report showed that a cystic or heterogeneous appearance, which depicts T2 high intensity on MRI, was associated with an increase in Antoni type B, and the prevalence of Antoni type B was correlated with a larger size compared with Antoni type A [23]. Therefore, this area may show higher proliferation and induce high APT SI. On the other hand, the alternation in tissue pH can affect APT SI [31,32].

APT SI can increase in tissues with increased pH because the amide proton exchange rate is base-catalyzed in the pH range. Higher pH increases the proton exchange rate, which increases the APT SI. Hemosiderin and xanthomatous cells were more frequently encountered in cystic VS [23]. However, it is unclear how these components contribute to pH. To the best of our knowledge, there have been no reports on pH about VS, and it is necessary to examine pH using surgical samples in the future.

In this study, benign schwannomas with T2 hyperintensity tended to show high APT SI, and this result may be specific to schwannomas. Our results suggest that APT SI may be able to distinguish between schwannoma and meningioma in the CPA.

### Limitations

This study had several limitations. First, our study included a small number of patients with schwannoma or meningioma and lacked an external validation cohort. Second, instead of evaluating entire tumors, the regions of interest (ROIs) were placed manually in the tumors where we thought the ROI most represented the tumor. We carefully attempted to exclude cystic portions, but some schwannomas with T2 hyperintensity had extensive cystic portions, and this may affect APT SI. We chose this method because tumors in the CPA occur near the skull, which is susceptible to field inhomogeneity and consequently may affect APT SI. However, the CPA seems to be very prone to image distortion with the echo-planning image (EPI) sequence used. This is because of the very narrow bandwidth in the phase-encoding direction with the EPI sequence. Magnetic field inhomogeneities cause substantial distortions of several voxels in extent, with severe piling up of signal from areas outside of the area of interest. Therefore, we were initially considering using APT images with single-shot fast spin echo to avoid field inhomogeneity. However, the single-shot fast spin echo sequence often failed at the CPA, likely because of the effect of bone, so we used the EPI sequence in this study. Finally, schwannomas with T2 high SI tend to show high APT SI. However, schwannomas with T2 low SI tend to show relatively low APT SI. Therefore, it may be difficult to distinguish between these and meningiomas.

## 4. Materials and Methods

### 4.1. Patients

From November 2020 to April 2022, 18 patients (4 males (22.2%) and 14 females (77.8%); mean age (standard deviation (SD)): 59.4 years (13.6)) underwent MRI with APT imaging in our hospital. All patients had undergone MRI in our hospital before the operation. All 18 resected samples were pathologically diagnosed.

Our institutional review board approved this study and waived the need for written informed consent because of the retrospective nature of the study. However, information that all patient data were used for research purposes was posted on the hospital’s homepage, giving patients the opportunity to refuse the use of their data.

### 4.2. MRI Protocol

The patients underwent MRI scans on a 3T scanner (Signa™ Architect, GE Healthcare, Milwaukee, WI, USA) with a 48-channel receiver array coil. Conventional MR images were acquired following the standard CPA brain tumor protocol in our hospital: (a) axial two-dimensional (2D) T2-weighted imaging (T2WI; repetition time (TR) = 3000 ms, echo time (TE) = 90 ms, field of view (FOV) = 18 × 20 cm^2^, slice thickness = 3 mm, matrix = 256 × 512, number of excitations (NEX) = 2); (b) axial DWI; TR = 5400 ms, TE = 73 ms, FOV = 22 × 25 cm^2^, slice thickness = 5 mm, matrix = 128 × 192, b-values = 0, 1000 s/mm^2^, NEX = 1) using single-shot EPI (SS-EPI); (c) fat suppression 3D T1-weighted imaging (T1WI; TR = 502 ms, TE = 17 ms, FOV = 22 × 25 cm^2^, flip angle = 90°, slice thickness = 1.2, matrix = 192 × 320, NEX = 2); and (d) fat suppression 3D contrast-enhanced T1WI (TR = 502 ms, TE = 17 ms, FOV = 16.0 × 17.7 cm^2^, flip angle = 90°, slice thickness = 1.2 mm, matrix = 256 × 256, NEX = 2). The total acquisition time for this protocol was 6 min and 18 s.

APT images were acquired before contrast administration using EPI with the following imaging parameters: field of view = 220 × 220 mm^2^, matrix = 128 × 128, spatial resolution = 1.7 × 1.7 mm^2^, slice thickness = 8.0 mm, TR/TE = 3000/26.6 ms, and number of slices = 1.

Twenty-nine saturation frequency offsets (7.0, 6.5, 6.0, 5.5, 5.0, 4.5, 4.0, 3.5, 3.0, 2.5, 2.0, 1.5, 1.0, 0.5, 0, −0.5, −1.0, −1.5, −2.0, −2.5, −3.0, −3.5, −4.0, −4.5, −5.0, −5.5, −6.0, −6.5, and −7.0 ppm) were used to attain sufficient a signal-to-noise ratio within the clinical time frame.

APT imaging consists of radiofrequency saturation (one pulse with a duration of 2000 ms and average B_1_ equivalent to a continuous radiofrequency power level of 2.0 µT, which has been widely adopted for clinical studies [33]).

Water frequency shift owing to field inhomogeneity was measured in a separate image acquired using the water-saturation shift referencing (WASSR) method with 11 offset frequencies ranging from −1.875 to 1.875 ppm, at intervals of 0.375 ppm, with one reference image acquired without a saturation RF pulse, resulting in a full Z-spectrum within the offset range. The WASSR image was acquired with a TR/TE of 3000/26.6 ms, RF saturation amplitude of 0.5 µT, and a total duration of 2000 ms, with a continuous wave. The total acquisition time for both APT and water-saturation shift reference images was 2 min and 9 s.

### 4.3. APT Image Processing

APT imaging data were analyzed in MATLAB (The MathWorks, Inc., Natick, MA, USA) on an MR scanner. The MTR_asym_ was also obtained. Using the shift-corrected data, the MTR_asym_ values at ±3.5 ppm with respect to water frequency were calculated as follows:(1)$${\mathrm{MTR}}_{\mathrm{asym}\;}\left(+3.5_{\mathrm{ppm}}\right)=\frac{S_{\mathrm{sat}}\left(-3.5_{\mathrm{ppm}}\right)-{\;S}_{\mathrm{sat}}\left(+3.5_{\mathrm{ppm}}\right)}{S_{0}}$$
where Ssat is the SI with selective imaging. S_0_ is the SI in the absence of RF for imaging SI normalization [34].

B_0_ inhomogeneity was corrected with a WASSR map [35] on a pixel-by-pixel basis.

### 4.4. Image Analysis

The APT images in this study were collected at preoperative MRI and independently evaluated by two neuroradiologists (with 12 and 36 years of experience in neuroradiology) who were blinded to the clinical data. A circular ROI was manually placed on a slice on raw APT imaging that exhibited similar contrast with T2WI for anatomical landmarks in the brain. The ROI was carefully placed to include the entire tumor to the extent possible but not to protrude from the tumor. This was necessary because the CPA seems to be very prone to image distortion with the EPI sequence, and the accurate location of the tumor is difficult to indicate. Moreover, we carefully attempted to exclude cystic portions in the tumor. The mean MTR_asym_ values of the two readers were used for analysis.

Conventional MR images were analyzed for MTD, T2 hyperintensity, peritumoral brain edema, irregular tumor margin, heterogeneous enhancement, capsular enhancement, “dural tail sign” (which shows thickening of the dura adjacent to an intracranial neoplasm on contrast-enhanced T1 MRI [36]), progress to the internal auditory canal, and eccentricity of tumor compared to nerve by the same two neuroradiologists. MTD was defined as the length of the long axis of the tumor on axial T2WI. T2 hyperintensity was defined as a higher SI of the tumor relative to gray matter. Irregular tumor margin was defined by the lobulated appearance of tumor margins on fat suppression 3D contrast-enhanced T1WI. The eccentricity of the tumor compared to the nerve was defined by acoustic nerve running along the inner margin of the tumor on fat suppression 3D contrast-enhanced T1WI.

If the diagnoses differed, the neuroradiologists reviewed the data to reach a consensus.

### 4.5. Pathological Diagnosis

Pathological diagnosis was made by a neuropathologist in our hospital following the WHO Histological Classification of Tumors of the Central Nervous System in 2016 [37].

### 4.6. Statistical Analysis

We used the D’Agostino–Pearson test to assess the normality of the data; non-normally distributed variables are presented as the median (range). Quantitative results are expressed as the mean ± SD or median (range).

Age, MTR_asym_ values, and MTD values were analyzed using the Wilcoxon signed-rank test. Sex, T2 hyperintensity, peritumoral brain edema, irregular tumor margin, heterogeneous enhancement, capsular enhancement, dural tail sign, and progress to the internal auditory canal were analyzed using the Chi-squared test. Results are expressed as sensitivity, specificity, and overall accuracy, with 95% CI calculated with the normal approximation method [38].

We created ROC curves and determined the threshold that led to the optimal values of probabilities in the schwannomas or meningiomas. The intersection of the ROC curve with the bisecting line at which sensitivity equaled specificity was considered the optimal threshold.

Interobserver agreement on MTR_asym_ in APT imaging and MTD in conventional MRI was evaluated by ICC, while that on conventional MRI features was evaluated by Cohen’s k coefficient. ICC and k values > 0.8 indicated excellent agreement, and >0.6 indicated good agreement.

For all tests, a two-sided *p*-value was used, and a *p*-value of <0.05 was considered statistically significant. Prism for Windows (version 8.3.0; GraphPad, San Diego, CA, USA) was used for all statistical analyses.

## 5. Conclusions

Schwannoma exhibited significantly higher APT SI than meningiomas. Schwannoma with T2 hyperintensity exhibited significantly higher APT SI than schwannoma with T2 low SI. Therefore, APT imaging can provide additional quantitative information for schwannoma, which may help distinguish schwannoma from meningioma.

## Figures and Tables

**Figure 1 ijms-23-10187-f001:**
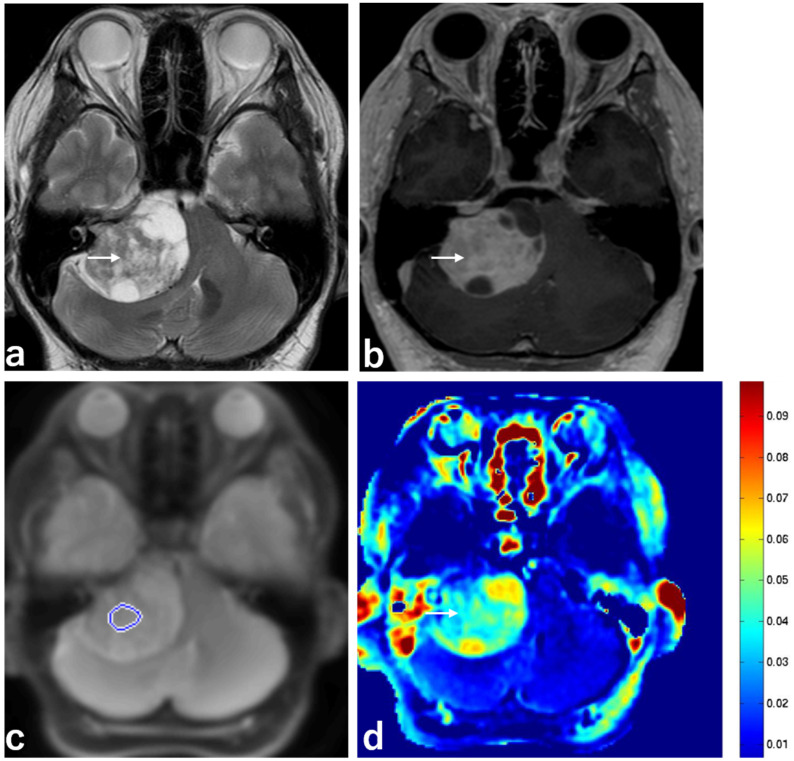
Imaging from a 43-year-old man with a schwannoma with T2 hyperintensity. (**a**) MRI, T2 sequence, axial section, showing heterogeneous high intensity in a schwannoma in the right cerebellopontine angle (CPA) (white arrow). (**b**) MRI, 3D contrast-enhanced T1 sequence, axial section, showing heterogeneous enhancement in a schwannoma in the right CPA (white arrow). (**c**) MRI, APT-CEST scout image, axial section, showing the ROI (blue circle) in a schwannoma in the right CPA. (**d**) MRI, APT-CEST sequence, axial section, showing high SI in a schwannoma in the right CPA (white arrow).

**Figure 2 ijms-23-10187-f002:**
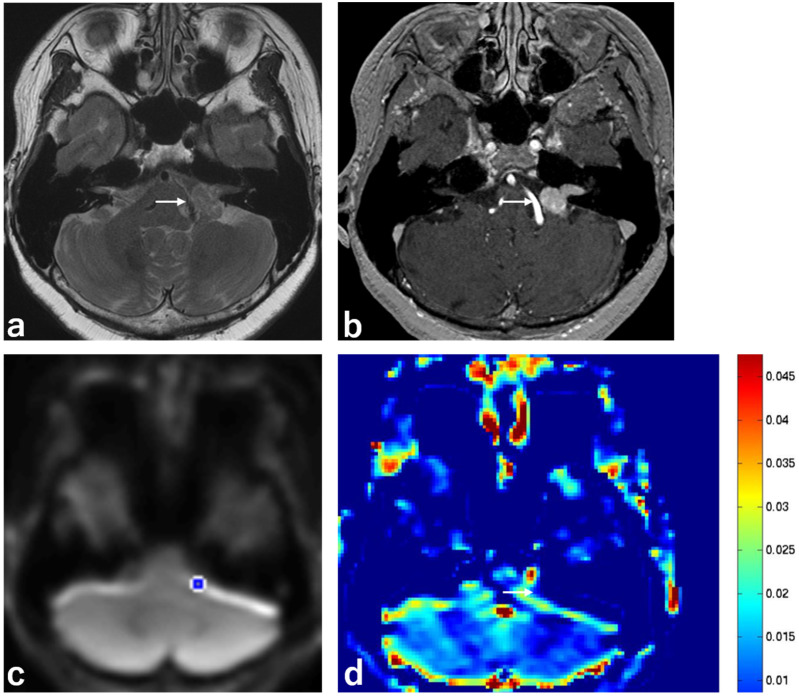
Imaging from a 66-year-old woman with a schwannoma with T2 low intensity. (**a**) MRI, T2 sequence, axial section, showing homogeneous low intensity in a schwannoma in the left CPA (white arrow). (**b**) MRI, 3D contrast-enhanced T1 sequence, axial section, showing homogeneous enhancement in a schwannoma in the left CPA (white arrow). (**c**) MRI, APT-CEST scout image, axial section, showing the ROI (blue circle) in a schwannoma in the left CPA. (**d**) MRI, APT-CEST sequence, axial section, showing low SI in a schwannoma in the left CPA (white arrow).

**Figure 3 ijms-23-10187-f003:**
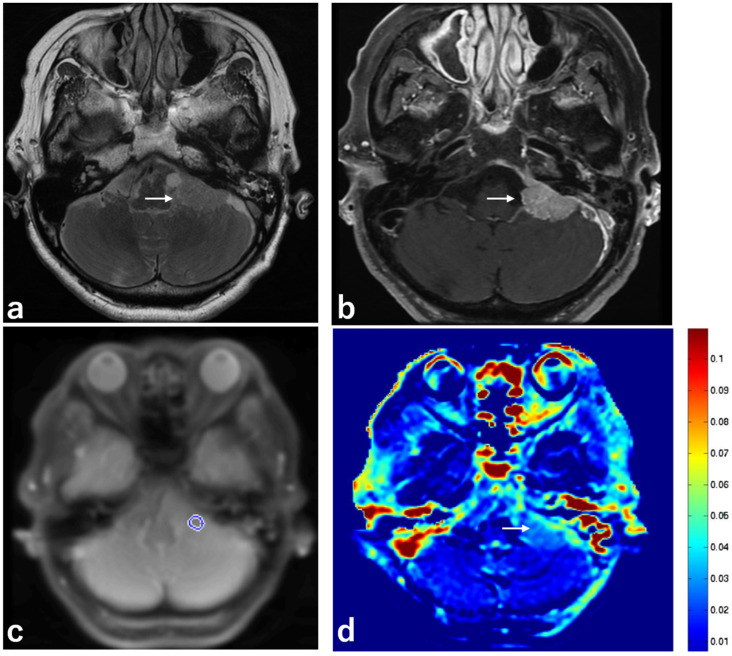
Imaging from a 78-year-old woman with a meningioma. (**a**) MRI, T2 sequence, axial section, showing homogeneous low intensity in a meningioma in the left CPA (white arrow). (**b**) MRI, 3D contrast-enhanced T1 sequence, axial section, showing homogeneous enhancement in a meningioma in the left CPA (white arrow). (**c**) MRI, APT-CEST scout image, axial section, showing the ROI (blue circle) in a meningioma in the left CPA. (**d**) MRI, APT-CEST sequence, axial section, showing low SI in a meningioma in the left CPA (white arrow).

**Figure 4 ijms-23-10187-f004:**
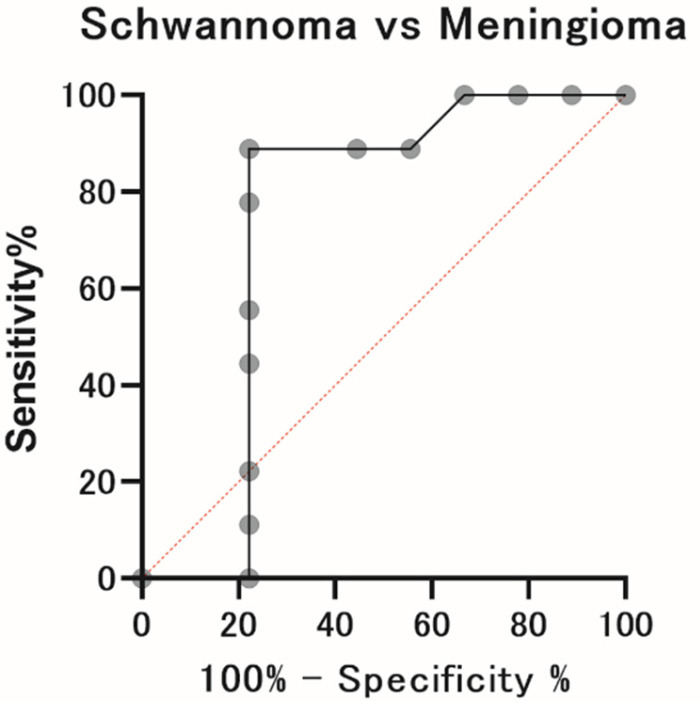
ROC analysis. ROC analyses demonstrated moderate discriminatory power for mean MTR_asym_ to differentiate between patients in the schwannoma group and meningioma group. When an AUC value of <0.024 was used as the threshold for diagnosis, the sensitivity, specificity, positive predictive value, and negative predictive value were 88.9%, 77.8%, 80.0%, and 87.5%, respectively.

**Table 1 ijms-23-10187-t001:** Comparison of demographic and clinical characteristics and imaging findings between patients with schwannomas and meningiomas.

Variable	Schwannoma (*n* = 9)	Meningioma (*n* = 9)	*p*
Age (years)	57.1 ± 12.1	61.7 ± 14.5	0.506
Male–female	2:7	2:7	1.000
Mean MTR_asym_	0.033 ± 0.012	0.021 ± 0.004	0.007
Mean MTD values (mm)	30.0 ± 7.2	36.6 ± 10.8	0.173
T2 hyperintensity	6 (66.7%)	2 (22.2%)	0.058
Peritumoral brain edema	2 (22.2%)	3 (33.3%)	0.599
Irregular tumor margin	2 (22.2%)	4 (44.4%)	0.137
Heterogeneous enhancement	6 (66.7%)	2 (22.2%)	0.058
Capsular enhancement	2 (22.2%)	3 (33.3%)	0.599
Dural tail sign	0 (0%)	6 (66.7%)	0.003
Progress to the internal auditory canal	6 (66.7%)	5 (55.6%)	0.629
Eccentricity of tumor compared to nerve	6 (66.7%)	2(22.2%)	0.058

MTR_asym_, magnetization transfer ratio asymmetry; MTD, maximum tumor diameter.

**Table 2 ijms-23-10187-t002:** Comparison of demographic and clinical characteristics and imaging findings between patients with schwannomas with T2 hyperintensity and low intensity.

Variable	Schwannoma with Hyperintensity (*n* = 6)	Schwannoma with Low Intensity (*n* = 3)	*p*
Age (years)	60.7 ± 10.8	50.0 ± 11.6	0.267
Male–female	1:5	1:2	0.571
Mean MTR_asym_	0.039 ± 0.006	0.021 ± 0.008	0.014
Mean MTD values (mm)	32.8 ± 6.9	24.4 ± 3.2	0.121
Peritumoral brain edema	1 (16.7%)	1 (33.3%)	0.571
Irregular tumor margin	1 (16.7%)	1 (33.3%)	0.571
Heterogeneous enhancement	6 (100%)	0 (0%)	0.003
Capsular enhancement	4 (66.7%)	1 (33.3%)	0.343
Dural tail sign	0 (0%)	0 (0%)	N.S.
Progress to the internal auditory canal	4 (66.7%)	2 (66.7%)	1.000

MTR_asym_, magnetization transfer ratio asymmetry; MTD, maximum tumor diameter; N.S., not significant.

## Data Availability

This study did not report any date.

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
