# Peer review of "Quantitative Chemical Exchange Saturation Transfer Imaging of Amide Proton Transfer Differentiates between Cerebellopontine Angle Schwannoma and Meningioma: Preliminary Results"

_ijms, 2022, doi:10.3390/ijms231710187_

Round 1

Reviewer 1 Report

This work retrospectively evaluated the potential of using amide proton transfer (APT)-chemical exchange saturation transfer (CEST) MRI to differentiate the vestibular schwannoma (VS) and meningioma at the cerebellopontine angle (CPA). MRI results from 18 patients, including 9 patients with schwannomas and 9 patients with meningioma, were analyzed. The APT-CEST results showed that schwannoma group has higher mean magnetization transfer ratio asymmetry (MTRasym) value/APT SI than the meningioma group, which may help distinguish schwannoma from meningioma. This work has explored the potential of APT-CEST application for clinical use. The manuscript is well presented, but still need some minor adjustments.

1.     Figure 1,2 and 3 all show: (a)T2 ; (b)3D contrast-enhanced T1; (c)ROI; (d) APT-CEST SI. Since same sequence and protocol has been used respectively, the contrast for T2 and T1 images should be similar or even identical between patients. For example, figure 1 (b) and figure 2 (b) showed very different scale for the image. Moreover, for the APT-CEST image (d in each figure), a color bar should be added to the map if it’s quantitative.

2.     P1, abstract, line 2, “followed by”.

3.     P1, Introduction, line 4, “Amide proton transfer (APT) imaging was the most often reported CEST technique…”, I think “was” should be “is”.

4.     In Table 1: are the “T2 hyperintensity” and “Heterogeneous enhancement” same patients? Same question for “Peritumoral brain edema” and “Capsular enhancement”. Need more details/explanations.

5.     P7, Discussion, 3rd paragraph: “The size of the VS is corrected to…”, should be “correlated”.

6.     P8, the first paragraph in this page, this sentence: “However, a previous report showed that a cystic or heterogeneous appearance, …” Can remove the word “However”. Because the report has not claimed anything contradict to this manuscript.

7.     P9, 5.2. MRI protocol: should show how long the total exam/scan time was.

8.     P10, second to the last paragraph in part  5.6, what is ICC in full name?

Author Response

Response to Reviewer:

We are very grateful for your positive comments.

  1. Figure 1,2 and 3 all show: (a)T2 ; (b)3D contrast-enhanced T1; (c)ROI; (d) APT-CEST SI. Since same sequence and protocol has been used respectively, the contrast for T2 and T1 images should be similar or even identical between patients. For example, figure 1 (b) and figure 2 (b) showed very different scale for the image. Moreover, for the APT-CEST image (d in each figure), a color bar should be added to the map if it’s quantitative.

Response:

Thank you for your recommendation. I changed figures.

  1. P1, abstract, line 2, “followed by”.

Response:

Thank you for your suggestion. I replaced “followed” with “followed by”.

  1. P1, Introduction, line 4, “Amide proton transfer (APT) imaging was the most often reported CEST technique…”, I think “was” should be “is”.

 Response:

Thank you for your suggestion. I replaced “was ” with “is”.

  1. In Table 1: are the “T2 hyperintensity” and “Heterogeneous enhancement” same patients? Same question for “Peritumoral brain edema” and “Capsular enhancement”. Need more details/explanations.

 Response:

Thank you for your suggestion. The patients with schwannoma or meningioma with “T2 hyperintensity” and “Heterogeneous enhancement” were same. The patients with schwannoma or meningioma with “Peritumoral brain edema” and “Capsular enhancement” were also same. I added these information (page 6, section 8-11) .

  1. P7, Discussion, 3rdparagraph: “The size of the VS is corrected to…”, should be “correlated”.

Response:

Thank you for your suggestion. I replaced “corrected ” with “correlated”.

  1. P8, the first paragraph in this page, this sentence: “However, a previous report showed that a cystic or heterogeneous appearance, …” Can remove the word “However”. Because the report has not claimed anything contradict to this manuscript.

Response:

Thank you for your recommendation. I removed the word “However”.

  1. P9, 5.2. MRI protocol: should show how long the total exam/scan time was.

Response:

Thank you for your recommendation. I added total scan time (page 25, section 6).

  1. P10, second to the last paragraph in part  5.6, what is ICC in full name?

Response:

Thank you for pointing out. ICC means intraclass coefficient. I spelled it in Result paragraph, interobserver agreement (page 19, section 5).

Reviewer 2 Report

In this study, the authors apply amide CEST imaging to distinguish CPA (cerebellopontine angle) schwannoma from CPA meningioma. This differential diagnosis is a classic challenge in neuroradiology, and the presented results may suggest that CEST imaging can aid this distinction, as higher MTR asymmetry values were seen in schwannomas. However, I think the study suffers from some limitations that may affect the results and the conclusions. As further explained below, given the present results, I still find it not-convincing that the CEST difference is due to a difference in these two types of tumors, and I think the authors have to perform additional analyses to prove it.

Major concerns (in order of relevance).

Major #1.

Tumors with “T2 hyperintensity” / cystic appearance.

This is the most relevant issue with the paper.

The authors report a difference in the occurrence of what they call “T2 hyperintensity” between schwannomas and meningiomas. In fact, T2 hyperintensity is more prevalent in schwannomas, and schwannomas are the tumor type with higher MTR asymmetry – see Table 1. In addition, they report that schwannomas with T2 hyperintensity have higher MTR asymmetry than schwannomas without T2 hyperintensity, and that schwannomas without T2 hyperintensity have a similar MTR asymmetry compared to meningiomas (0.021) – see Table 2. Finally, the wording “T2 hyperintensity” is probably misleading because in the representative case (Figure 1) such T2 hyperintensity corresponds to a cystic area, and not to a T2-hyperintense solid area (and the ROI was placed within the cyst).

All these considerations, taken together, lead me to hypothesize that the CEST difference may be driven by whether the ROI is placed in a cystic area (high MTR asymmetry) or in a solid area (low MTR asymmetry), rather than driven by the difference between schwannoma and meningioma.

In fact, as already stated, schwannomas without T2 hyperintensity had MTR asymmetry 0.021, just like meningiomas.

A.    My first set of questions in this regard would be: how many of these T2 hyperintense cases were cystic, and how many instead showed a T2 hyperintensity within the solid tissue? Were the ROIs ever placed in the solid tissue? In other words, a high MTR asymmetry (~0.045) was only found in cystic areas?

B.    I think the authors should run additional analyses to prove that the CEST difference is not due to the ROI being placed in the cystic areas. A possible solution is to enroll an additional cohort of non-cystic schwannomas and show high MTR asymmetry in such patients. Alternatively, the authors may revise these 18 patients and perform an additional analysis by carefully placing the ROIs in the solid tissue rather than the cystic parts (unlike they did in figure 1).

Major #2.

ROI placement method.

I think it is ok to sample MTR asymmetry values with a 2D planar ROI instead of 3D segmentation. However (see also Major#1), placing the ROI based on the CEST scout image (panel C of figures 1,2,3) is not a great method because the reader cannot be aware whether the ROI will end up in enhancing tissue, in cystic areas, and so on. That is, the quantitative comparisons may end up comparing apple and oranges. As I stated in Major#1, the authors may have compared CEST from cystic areas in schwannomas vs CEST from solid areas in meningiomas.

In case the authors are willing to follow my advice in Major#1B, I would also recommend they perform image registration between conventional MRI (cMRI) and CEST, so that the reader is aware where they are placing the ROI. Since the authors are working in Matlab already, image registration should be easy to include.

Major #3.

Amide CEST interpretation.

The authors state that the source of APT signal intensity is unknown (introduction) and that may be influenced by tumor cellularity (introduction and discussion). As far as I know, there is extensive literature explaining how amide CEST contrast is mainly used as a proxy of acidity (therefore sometimes called “pH-imaging”) [see Cho, Hagiwara et al. 2022 NMR in Biomedicine, and also Goldenberg and Pagel 2019 NMR in Biomedicine].

I was very surprised when I did not found any reference to pH or acidity in the whole paper. This should be included and used to discuss the data: maybe schwannomas are more acidic? Why would they be? Maybe the cystic parts are more acidic? (see Major#1).

Major #4.

Eccentricity.

A classic cMRI sign that helps with schwannoma vs meningioma differential diagnosis is that schwannomas tend to be eccentric compared to the nerve [see for instance Kutz Barnett et al, Skull Base 2009]. I wonder why the authors did not include this classic cMRI feature in their evaluation, since I don’t see it either in table 1 or in the text.

Strengths of the study

1.     Assessment of inter-reader agreement.

2.     Method for including patients: all CPA lesions between November 2020 to April 2022. To this regard, I was wondering: do the authors have an explanation about the ratio schannomas:meningiomas being 1:1 in their cohort, whereas it is supposed to be ~4:1 in the general population?

3.     Applying an advanced technique to schwannomas for the first time

Author Response

Response to Reviewer:

We are very grateful for your positive comments.

Major concerns (in order of relevance).

Major #1.

Tumors with “T2 hyperintensity” / cystic appearance.

This is the most relevant issue with the paper.

The authors report a difference in the occurrence of what they call “T2 hyperintensity” between schwannomas and meningiomas. In fact, T2 hyperintensity is more prevalent in schwannomas, and schwannomas are the tumor type with higher MTR asymmetry – see Table 1. In addition, they report that schwannomas with T2 hyperintensity have higher MTR asymmetry than schwannomas without T2 hyperintensity, and that schwannomas without T2 hyperintensity have a similar MTR asymmetry compared to meningiomas (0.021) – see Table 2. Finally, the wording “T2 hyperintensity” is probably misleading because in the representative case (Figure 1) such T2 hyperintensity corresponds to a cystic area, and not to a T2-hyperintense solid area (and the ROI was placed within the cyst).

All these considerations, taken together, lead me to hypothesize that the CEST difference may be driven by whether the ROI is placed in a cystic area (high MTR asymmetry) or in a solid area (low MTR asymmetry), rather than driven by the difference between schwannoma and meningioma.

In fact, as already stated, schwannomas without T2 hyperintensity had MTR asymmetry 0.021, just like meningiomas.

Response:

Thank you for your suggestion. I think “T2 hyperintensity” is confusing. However, T2 hyperintensity area may include both cystic area and Antoni type B pattern area.

  1. My first set of questions in this regard would be: how many of these T2 hyperintense cases were cystic, and how many instead showed a T2 hyperintensity within the solid tissue? Were the ROIs ever placed in the solid tissue? In other words, a high MTR asymmetry (~0.045) was only found in cystic areas?

Response:

We reviewed schwannomas with T2 hyperintensity in this study. T2 hyper intensity areas in all six lesions were predominantly cystic components.  However, the Antoni B region also has a poor enhancing effect, so there is a possibility that it is mixed. Pathologically, cystic VS were predominantly mixed (Antoni A and B) as previous literature.

  1. I think the authors should run additional analyses to prove that the CEST difference is not due to the ROI being placed in the cystic areas. A possible solution is to enroll an additional cohort of non-cystic schwannomas and show high MTR asymmetry in such patients. Alternatively, the authors may revise these 18 patients and perform an additional analysis by carefully placing the ROIs in the solid tissue rather than the cystic parts (unlike they did in figure 1).

Response:

All 6 schwannomas with T2 hyperintensity in this study consisted of mostly poorly enhanced areas of T2 hyperintensity and a few areas of enhancement presumed to be antoni A areas. Therefore, T2 hyperintensity area may be considered as cysts or antoni B areas, and there were no solid parts showing T2 hyperintensity.

Major #2.

ROI placement method.

I think it is ok to sample MTR asymmetry values with a 2D planar ROI instead of 3D segmentation. However (see also Major#1), placing the ROI based on the CEST scout image (panel C of figures 1,2,3) is not a great method because the reader cannot be aware whether the ROI will end up in enhancing tissue, in cystic areas, and so on. That is, the quantitative comparisons may end up comparing apple and oranges. As I stated in Major#1, the authors may have compared CEST from cystic areas in schwannomas vs CEST from solid areas in meningiomas.

In case the authors are willing to follow my advice in Major#1B, I would also recommend they perform image registration between conventional MRI (cMRI) and CEST, so that the reader is aware where they are placing the ROI. Since the authors are working in Matlab already, image registration should be easy to include.

Response:

Thank you for your suggestion. In fact, all 6 schwannomas with T2 hyperintensity in this study had almost no solid components (enhanced areas). Therefore, tumors are thought to be dominated by cysts (or Antoni B regions), and we did not compare CEST from T2 hyperintensity solid areas in schwannomas vs CEST from solid areas in meningiomas.

However, we believe that this T2 hyperintensity area with poor enhancement effect which shows high APT SI may be characteristic of schwannoma. Moreover, proof of cyst or Antoni B regions in schwannoma requires contrast agent on MRI. On the other hand, APT-CEST image does not require exogenous contrast and is safe in patients with renal failure or with intolerance to contrast media. Our results may be useful in diagnosing schwannoma.

Major #3.

Amide CEST interpretation.

The authors state that the source of APT signal intensity is unknown (introduction) and that may be influenced by tumor cellularity (introduction and discussion). As far as I know, there is extensive literature explaining how amide CEST contrast is mainly used as a proxy of acidity (therefore sometimes called “pH-imaging”) [see Cho, Hagiwara et al. 2022 NMR in Biomedicine, and also Goldenberg and Pagel 2019 NMR in Biomedicine].

I was very surprised when I did not found any reference to pH or acidity in the whole paper. This should be included and used to discuss the data: maybe schwannomas are more acidic? Why would they be? Maybe the cystic parts are more acidic? (see Major#1).

Response:

Thank you for your recommendation. I added discussion about pH  (page 22, section 8-15).

Major #4.

Eccentricity.

A classic cMRI sign that helps with schwannoma vs meningioma differential diagnosis is that schwannomas tend to be eccentric compared to the nerve [see for instance Kutz Barnett et al, Skull Base 2009]. I wonder why the authors did not include this classic cMRI feature in their evaluation, since I don’t see it either in table 1 or in the text.

Response:

Thank you for your recommendation. I added “eccentricity of tumor compared to nerve” in table 1.

Round 2

Reviewer 2 Report

Here are my further comments (in red), corresponding to the Authors’ answers.

Overall, I think that the main issues with the paper have not been addressed.

The results do not really prove the added value of CEST for differential diagnosis, compared to conventional imaging.

Major concerns (in order of relevance).

Major #1.

Tumors with “T2 hyperintensity” / cystic appearance.

This is the most relevant issue with the paper.

The authors report a difference in the occurrence of what they call “T2 hyperintensity” between schwannomas and meningiomas. In fact, T2 hyperintensity is more prevalent in schwannomas, and schwannomas are the tumor type with higher MTR asymmetry – see Table 1. In addition, they report that schwannomas with T2 hyperintensity have higher MTR asymmetry than schwannomas without T2 hyperintensity, and that schwannomas without T2 hyperintensity have a similar MTR asymmetry compared to meningiomas (0.021) – see Table 2. Finally, the wording “T2 hyperintensity” is probably misleading because in the representative case (Figure 1) such T2 hyperintensity corresponds to a cystic area, and not to a T2-hyperintense solid area (and the ROI was placed within the cyst).

All these considerations, taken together, lead me to hypothesize that the CEST difference may be driven by whether the ROI is placed in a cystic area (high MTR asymmetry) or in a solid area (low MTR asymmetry), rather than driven by the difference between schwannoma and meningioma.

In fact, as already stated, schwannomas without T2 hyperintensity had MTR asymmetry 0.021, just like meningiomas.

Response:

Thank you for your suggestion. I think “T2 hyperintensity” is confusing. However, T2 hyperintensity area may include both cystic area and Antoni type B pattern area.

A.      My first set of questions in this regard would be: how many of these T2 hyperintense cases were cystic, and how many instead showed a T2 hyperintensity within the solid tissue? Were the ROIs ever placed in the solid tissue? In other words, a high MTR asymmetry (~0.045) was only found in cystic areas?

Response:

We reviewed schwannomas with T2 hyperintensity in this study. T2 hyper intensity areas in all six lesions were predominantly cystic components.  However, the Antoni B region also has a poor enhancing effect, so there is a possibility that it is mixed. Pathologically, cystic VS were predominantly mixed (Antoni A and B) as previous literature.

B.       I think the authors should run additional analyses to prove that the CEST difference is not due to the ROI being placed in the cystic areas. A possible solution is to enroll an additional cohort of non-cystic schwannomas and show high MTR asymmetry in such patients. Alternatively, the authors may revise these 18 patients and perform an additional analysis by carefully placing the ROIs in the solid tissue rather than the cystic parts (unlike they did in figure 1).

Response:

All 6 schwannomas with T2 hyperintensity in this study consisted of mostly poorly enhanced areas of T2 hyperintensity and a few areas of enhancement presumed to be antoni A areas. Therefore, T2 hyperintensity area may be considered as cysts or antoni B areas, and there were no solid parts showing T2 hyperintensity.

The authors did not perform any additional analyses to address this problem. Therefore, the results still suggest that the CEST difference is seen between cyst and tissue, rather than between schwannomas and meningiomas: again, non-cystic schwannomas (n=3) showed the same MTR asymmetry as meningiomas (0.021 – see table 2).

The question is: what happens when the schwannoma is not cystic? Can CEST still help in the differential diagnosis? As already stated, the study only includes 3 non-cystic schwannomas, which show the same CEST features as meningiomas. So, at the present moment, the answer is no, CEST does not help unless the lesion is cystic.

So, CEST is pretty much useless because assessing the presence of cystic components would be enough for differential diagnosis, without implementing CEST which is technically demanding and requires postprocessing.

Major #2.

ROI placement method.

I think it is ok to sample MTR asymmetry values with a 2D planar ROI instead of 3D segmentation. However (see also Major#1), placing the ROI based on the CEST scout image (panel C of figures 1,2,3) is not a great method because the reader cannot be aware whether the ROI will end up in enhancing tissue, in cystic areas, and so on. That is, the quantitative comparisons may end up comparing apple and oranges. As I stated in Major#1, the authors may have compared CEST from cystic areas in schwannomas vs CEST from solid areas in meningiomas.

In case the authors are willing to follow my advice in Major#1B, I would also recommend they perform image registration between conventional MRI (cMRI) and CEST, so that the reader is aware where they are placing the ROI. Since the authors are working in Matlab already, image registration should be easy to include.

Response:

Thank you for your suggestion. In fact, all 6 schwannomas with T2 hyperintensity in this study had almost no solid components (enhanced areas). Therefore, tumors are thought to be dominated by cysts (or Antoni B regions), and we did not compare CEST from T2 hyperintensity solid areas in schwannomas vs CEST from solid areas in meningiomas.

However, we believe that this T2 hyperintensity area with poor enhancement effect which shows high APT SI may be characteristic of schwannoma. Moreover, proof of cyst or Antoni B regions in schwannoma requires contrast agent on MRI. On the other hand, APT-CEST image does not require exogenous contrast and is safe in patients with renal failure or with intolerance to contrast media. Our results may be useful in diagnosing schwannoma.

Related to Major #1, the authors are admitting that the ROI placement was in the cystic components (when present). Therefore, CEST differences are driven by the presence of cystic components and assessing T2 hyperintensity is pretty much enough for differential diagnosis.

Regarding the need for contrast agent injection for the assessment of cystic components, I disagree: T2 weighted imaging, FLAIR, and DWI/ADC are more than enough.

In addition, in the presence of a CPA lesion, contras agent injection is fundamental to assess the extent of the lesion for treatment purposes. This evaluation is even more important than knowing in advance whether the lesion is a schwannoma or a meningioma. In addition, unlike iodine, gadolinium very rarely causes allergies and is feasible also for patients with a relatively low renal function.

Major #3.

Amide CEST interpretation.

The authors state that the source of APT signal intensity is unknown (introduction) and that may be influenced by tumor cellularity (introduction and discussion). As far as I know, there is extensive literature explaining how amide CEST contrast is mainly used as a proxy of acidity (therefore sometimes called “pH-imaging”) [see Cho, Hagiwara et al. 2022 NMR in Biomedicine, and also Goldenberg and Pagel 2019 NMR in Biomedicine].

I was very surprised when I did not found any reference to pH or acidity in the whole paper. This should be included and used to discuss the data: maybe schwannomas are more acidic? Why would they be? Maybe the cystic parts are more acidic? (see Major#1).

Response:

Thank you for your recommendation. I added discussion about pH  (page 22, section 8-15).

Thanks for adding it.

Major #4.

Eccentricity.

A classic cMRI sign that helps with schwannoma vs meningioma differential diagnosis is that schwannomas tend to be eccentric compared to the nerve [see for instance Kutz Barnett et al, Skull Base 2009]. I wonder why the authors did not include this classic cMRI feature in their evaluation, since I don’t see it either in table 1 or in the text.

Response:

Thank you for your recommendation. I added “eccentricity of tumor compared to nerve” in table 1.

 Thanks for adding it.

Author Response

 Here are my further comments (in red), corresponding to the Authors’ answers.

Overall, I think that the main issues with the paper have not been addressed.

The results do not really prove the added value of CEST for differential diagnosis, compared to conventional imaging.

Response:

I am sorry. As you said, I think that these results have not proven the utility of CEST for differential diagnosis. I reconsidered this study.

Major concerns (in order of relevance).

Major #1.

Tumors with “T2 hyperintensity” / cystic appearance.

This is the most relevant issue with the paper.

The authors report a difference in the occurrence of what they call “T2 hyperintensity” between schwannomas and meningiomas. In fact, T2 hyperintensity is more prevalent in schwannomas, and schwannomas are the tumor type with higher MTR asymmetry – see Table 1. In addition, they report that schwannomas with T2 hyperintensity have higher MTR asymmetry than schwannomas without T2 hyperintensity, and that schwannomas without T2 hyperintensity have a similar MTR asymmetry compared to meningiomas (0.021) – see Table 2. Finally, the wording “T2 hyperintensity” is probably misleading because in the representative case (Figure 1) such T2 hyperintensity corresponds to a cystic area, and not to a T2-hyperintense solid area (and the ROI was placed within the cyst).

All these considerations, taken together, lead me to hypothesize that the CEST difference may be driven by whether the ROI is placed in a cystic area (high MTR asymmetry) or in a solid area (low MTR asymmetry), rather than driven by the difference between schwannoma and meningioma.

In fact, as already stated, schwannomas without T2 hyperintensity had MTR asymmetry 0.021, just like meningiomas.

Response:

Thank you for your suggestion. I think “T2 hyperintensity” is confusing. However, T2 hyperintensity area may include both cystic area and Antoni type B pattern area.

  1. My first set of questions in this regard would be: how many of these T2 hyperintense cases were cystic, and how many instead showed a T2 hyperintensity within the solid tissue? Were the ROIs ever placed in the solid tissue? In other words, a high MTR asymmetry (~0.045) was only found in cystic areas?

Response:

We reviewed schwannomas with T2 hyperintensity in this study. T2 hyper intensity areas in all six lesions were predominantly cystic components.  However, the Antoni B region also has a poor enhancing effect, so there is a possibility that it is mixed. Pathologically, cystic VS were predominantly mixed (Antoni A and B) as previous literature.

  1. I think the authors should run additional analyses to prove that the CEST difference is not due to the ROI being placed in the cystic areas. A possible solution is to enroll an additional cohort of non-cystic schwannomas and show high MTR asymmetry in such patients. Alternatively, the authors may revise these 18 patients and perform an additional analysis by carefully placing the ROIs in the solid tissue rather than the cystic parts (unlike they did in figure 1).

Response:

All 6 schwannomas with T2 hyperintensity in this study consisted of mostly poorly enhanced areas of T2 hyperintensity and a few areas of enhancement presumed to be antoni A areas. Therefore, T2 hyperintensity area may be considered as cysts or antoni B areas, and there were no solid parts showing T2 hyperintensity.

The authors did not perform any additional analyses to address this problem. Therefore, the results still suggest that the CEST difference is seen between cyst and tissue, rather than between schwannomas and meningiomas: again, non-cystic schwannomas (n=3) showed the same MTR asymmetry as meningiomas (0.021 – see table 2).

The question is: what happens when the schwannoma is not cystic? Can CEST still help in the differential diagnosis? As already stated, the study only includes 3 non-cystic schwannomas, which show the same CEST features as meningiomas. So, at the present moment, the answer is no, CEST does not help unless the lesion is cystic.

So, CEST is pretty much useless because assessing the presence of cystic components would be enough for differential diagnosis, without implementing CEST which is technically demanding and requires postprocessing.

Response:

Yes, I thought that pointing out the cystic component of schwannoma on CEST would not be very useful in differentiating it from meningioma. Therefore, we reviewed 6 schwannoma with T2 hyperintensity. We placed the ROI on a solid portion with T2  hyperintensity to avoid containing cystic components as much as possible. As a result, APT SI showed hyperintensity, although not as much as the cyst component.

The results of this study also showed that APT SI in schwannoma had a significantly higher signal than meningioma. APT SI in schwannoma with T2 hyperintensity also had a significantly higher than schwannoma with T2 low intensity. I changed figure 1 to be more representative. However, some schwannomas with T2 hyperintensity had extensive cystic portions, and this may affect APT SI.  I added this to the limitation paragraph.

Major #2.

ROI placement method.

I think it is ok to sample MTR asymmetry values with a 2D planar ROI instead of 3D segmentation. However (see also Major#1), placing the ROI based on the CEST scout image (panel C of figures 1,2,3) is not a great method because the reader cannot be aware whether the ROI will end up in enhancing tissue, in cystic areas, and so on. That is, the quantitative comparisons may end up comparing apple and oranges. As I stated in Major#1, the authors may have compared CEST from cystic areas in schwannomas vs CEST from solid areas in meningiomas.

In case the authors are willing to follow my advice in Major#1B, I would also recommend they perform image registration between conventional MRI (cMRI) and CEST, so that the reader is aware where they are placing the ROI. Since the authors are working in Matlab already, image registration should be easy to include.

Response:

Thank you for your suggestion. In fact, all 6 schwannomas with T2 hyperintensity in this study had almost no solid components (enhanced areas). Therefore, tumors are thought to be dominated by cysts (or Antoni B regions), and we did not compare CEST from T2 hyperintensity solid areas in schwannomas vs CEST from solid areas in meningiomas.

However, we believe that this T2 hyperintensity area with poor enhancement effect which shows high APT SI may be characteristic of schwannoma. Moreover, proof of cyst or Antoni B regions in schwannoma requires contrast agent on MRI. On the other hand, APT-CEST image does not require exogenous contrast and is safe in patients with renal failure or with intolerance to contrast media. Our results may be useful in diagnosing schwannoma.

Related to Major #1, the authors are admitting that the ROI placement was in the cystic components (when present). Therefore, CEST differences are driven by the presence of cystic components and assessing T2 hyperintensity is pretty much enough for differential diagnosis.

Regarding the need for contrast agent injection for the assessment of cystic components, I disagree: T2 weighted imaging, FLAIR, and DWI/ADC are more than enough.

In addition, in the presence of a CPA lesion, contras agent injection is fundamental to assess the extent of the lesion for treatment purposes. This evaluation is even more important than knowing in advance whether the lesion is a schwannoma or a meningioma. In addition, unlike iodine, gadolinium very rarely causes allergies and is feasible also for patients with a relatively low renal function.

Response:

I think you are right. I thought that it would be useful to point out the cyst component at CEST. However, as you said, I think T2WI, FLAIR, or DWI/ADC would suffice. Contrast-enhanced MRI is easier to use than contrast-enhanced CT, and its use is essential in evaluating tumors at CPA.

Round 3

Reviewer 2 Report

I thank the authors for addressing my comments and for re-analysing the cystic lesions in order to report MTR asymmetry values corresponding to the solid tissue portions. I think now the manuscript is more valuable because the main message is that CEST metrics are different between Schwannoma and Meningioma solid tissue (non-cystic) portions.

Minor comment: table 1 and table 2 are not complying with the journal template.